# The topology of the reaction stereo-dynamics in chemi-ionizations

Stefano Falcinelli [1✉], Franco Vecchiocattivi [1] & Fernando Pirani [1,2]

Details on the stereo-dynamic topology of chemi-ionizations highlight the role of the centrifugal barrier of colliding reactants: it acts as a selector of the orbital quantum number effective for reaction in a state-to-state treatment. Here, an accurate internally consistent formulation of the Optical interaction potentials, obtained by the combined analysis of scattering and spectroscopic experimental findings, casts light on structure, energy and angular momentum couplings of the precursor (pre-reactive) state controlling the stereo-dynamics of prototypical chemi-ionization reactions. The closest approach (turning point) of reagents, is found to control the relative weight of two different reaction mechanisms: (i) A direct mechanism stimulated by exchange chemical forces mainly acting at short separation distances and high collision energy; (ii) An indirect mechanism, caused by the combination of weak chemical and physical forces dominant at larger distances, mainly probed at low collision energy, that can be triggered by a virtual photon exchange between reagents.

[1] Department of Civil and Environmental Engineering, University of Perugia, Via G. Duranti 93, 06125 Perugia, Italy. [2] Department of Chemistry, Biology and Biotechnologies, University of Perugia, Via Elce di Sotto 8, 06123 Perugia, Italy. ✉email: stefano.falcinelli@unipg.it

The aim of the present work is to characterize basic details on the topology of the stereo-dynamics controlling chemi-ionization reactions, an important class of bimolecular processes, triggered by electronically excited species, occurring in flames, plasmas, planetary atmospheres, and interstellar medium.

Since long time the characterization of the stereo-dynamics of elementary chemical-physical processes represents a hot topic of research (see for instance refs. [1–13] and references therein). Presently, this target can be pursued by investigating in detail, under controlled conditions, some prototypical cases as those presented in this study. In particular, the focus is on barrier-less chemi-ionization reactions in order to provide unique-direct information on basic quantities determining the topology of their stereo-dynamics. The precursor state (PS), formed by collisions of reagents, plays here an important role being coincident with the reaction transition state. All the features of such a state, as well as its structure and stability, are directly or indirectly controlled by intermolecular forces operative in each collision complex.

Chemi-ionization reactions (CHEMI) are bimolecular processes involving as reagents a highly energetic $X^*$ species, electronically excited in a metastable state, interacting with another partner atomic or molecular M (see for instance refs. [7,8,13–30] and references therein). Therefore, CHEMI can be so schematized:

$$X^* + M \rightarrow [X \cdots M]^* \rightarrow [X \cdots M]^+ + e^- \rightarrow \text{ion products} \quad (1)$$

Where $[X \cdots M]^+$ is an intermediate ionic collision complex formed by a spontaneous autoionization of the $[X \cdots M]^*$ neutral excited PS which is the reaction transition state (TS) of such processes. Recently, it has been shown that the $[X \cdots M]^*$ TS can evolve towards the formation of the final ions via two microscopic reaction mechanisms[23–25]: (i) An exchange mechanism (oxidative-chemical), hereinafter referred to as direct mechanism (DM). It is triggered by strong chemical forces that act mainly at short separation distances, inducing an electron transfer between the reactants through a prototype oxidation process; (ii) A radiative (optical-physical) mechanism, henceforth referred to in the text as an indirect mechanism (IM). It is caused by the combination of weak chemical and physical forces dominant at intermediate/larger distances and which provokes an electron ejection via a pure photoionization event determined by the exchange of a virtual photon between the reacting partners[29]. The occurrence of the two mechanisms has been suggested several years ago[28,29] (see also ref. [30] and references therein) but only recently the nature and selectivity of intermolecular forces involved have been completely addressed[23–25].

It is relevant to note that because of an external electronic cloud polarization, the PS $[X \cdots M]^*$ formed at short separation distances tends to assume the configuration of an internal ionic core surrounded by a Rydberg electron. Therefore, DM can be considered as a particular case of ion chemistry, triggered by non-resonant charge transfer coupling entrance and exit channels[25].

Usually, $X^*$ can be formed by the collision of X, in its ground electronic state, with electrons or cosmic rays, and it exhibits a lifetime sufficiently long to permit, in a bulk, several collisions with M, until it disappears for inelastic and/or reactive collisions. CHEMI involves $X^*$ and M as neutral reagents and the associated $XM^+$, the parent $M^+$ (with its fragmentation species) plus electrons as ionic products. They are barrier-less processes with stereo-dynamics controlled by a PS, in this case, coincident with the reaction TS. Therefore, they represent prototypes of elementary processes triggered by pronounced electronic rearrangements occurring in PS.

CHEMI are of great interest for fundamental and applied research[7,8,13–30] since they participate to the balance of elementary processes in several environments, where $X^*$ easily forms.

They happen both at very low-temperature T (under conditions of the cold chemistry)[18,21], as in interstellar medium, at intermediate T values, as in planetary atmospheres[25], and at high T, as in flames and plasmas (see refs. [24,25] and references therein). CHEMI has been the target of several experimental and theoretical investigations[13–32]. Most of the experimental studies, carried out in the gas phase with the molecular beam (MB) technique, resolved the reactivity dependence on the stereo-dynamics of single collision events.

As mentioned above, rationalizing a huge number of experimental data from various laboratories on CHEMI promoted by $Ne^*(^3P_{2,0})$ metastable atoms, we were able to generalize[23–25] that, CHEMI occur via the DM, triggered by strong chemical forces, and/or via IM, controlled by the balance of weak intermolecular forces, having both chemical and physical nature. As stressed above, DM confines CHEMI in the framework of elementary exothermic oxidation reactions, while IM, usually stimulated by a virtual photon exchange between reagents, makes the reaction like an elementary photo-ionization process[24,25,30]. The relative role of DM/IM is found to be directly dependent on the separation distance R mainly probed by reagents in the entrance channels[24,25].

The present study exploits an accurate formulation of the intermolecular interaction potentials, previously obtained[23–25,33] and suggested by the complex phenomenology of open-shell P atoms, investigated by advanced experimental and theoretical methods, by the behaviour of ion-neutral systems, coupled by charge transfer, and by the spectroscopic properties of excimers. The obtained formulation has been tested on experimental findings of CHEMI, investigated in our and other laboratories by coupling scattering and spectroscopic techniques[7,8,23–25,34,35]. In order to emphasize innovative aspects of the reaction stereo-dynamics, we have found it convenient to refer to particular geometries of PS (or pre-reactive state), that open specific reaction channels. However, in the analysis and tests on the experimental findings, the full space of the relative geometries of reagents has been considered. In addition, in the case of $NH_3$ CHEMI the two considered geometries are the most relevant ones promoting, within selected angular cones, the formation of $NH_3^+$ ionic product in the ground (X) and in the first excited (A) electronic states[7,8,35]. The acceptance of angular cones is discussed in detail in ref. [8]. Therefore, the selection of individual reaction channels, triggered by specific cuts of the multi-dimensional interaction potential has been useful to properly address the following aspects of general interest:

How structure and energetic of PS (or TS), formed via collision of reagents, depending on the critical balance between the anisotropic interaction potential V, the collision energy, $E_{coll}$, and the orbital angular momentum quantum number, $\ell$, accompanying any scattering event. In a classical picture $\ell$ corresponds to the impact parameter b.

The role of the closest approach distance $R_c$ (turning point), associated to each collision event, that limits the range of intermolecular distances R affecting the process[7,8,30]. Around $R_c$ the valence orbital overlap is highest, the system inverts its relative motion, the radial velocity tends to zero and here the system spends most part of its collision time. Therefore, for each $\ell/b$ value, CHEMI has the highest probability to occur in the neighbourhood of $R_c$.

The change of the relative role of DM and IM that provides basic details on the dependence of the microscopic evolution on $E_{coll}$ and $\ell/b$.

The limitations of theoretical models, exclusively based on the capture by long-range attractions, that are omitting the selectivity of phenomena triggered by forces emerging at shorter distances.

The criteria to be adopted to properly assess the orientation degree of polar reagents in the electric field gradient associated to the anisotropic intermolecular forces, controlling the collision dynamics.

Thanks to the above, to the best of our knowledge, we are able to perform the first original attempt to point out how the chemical reactivity depends on the angular momentum quantum number of the intermediate collision complex (precursor or pre-reactive state) leading to the reaction. In particular, we are able to clarify the way in which the centrifugal barrier of the colliding reagents strongly acts as a sort of selector of the orbital quantum number effective for the reaction. In our knowledge, this is the first attempt to depict the topology of the stereo-dynamics of a chemical reaction by a state-to-state treatment.

The results of present study, casting light on basic details of the topology of CHEMI stereo-dynamics, appear of great interest for the present as for many other processes, where they are operative but are usually obscured by other effects. In several cases such details are difficult to obtain directly, being mixed with other effects, as the averaging role by many body and multiple collision events, the masking contributions of the solvent, present in several reaction environments. Furthermore, obtained results suggest some limitations of the so-called capture models, usually exploited to describe many other barrier-less processes, like ion–molecule reactions, occurring in interstellar medium, planetary atmospheres, and plasmas.

Moreover, the different structure of the most stable PS respect to TS opposing the lowest energy barrier to reagents-products passage, makes the rationalization more difficult. Note that any speculation on CHEMI stereo-dynamics presented in this paper exploits a semi-classical treatment of the collision dynamics[25,30] whose general aspects are given in the Supplementary Methods section of SI. Such treatment is quantitative for collision events occurring with relative kinetic energy confined in the thermal and hyper-thermal range of values and provides results that, at a semi-quantitative level, are proper to emphasize important selectivity of the scattering also under sub-thermal conditions. The treatment becomes less accurate when resonances, due to quantum effects, are observed, that effectively manifest when light reagents collide at low kinetic energy. This is confirmed by the observation of resonances in $He^*(^3S_1) + NH_3$ CHEMI[20,36] due to orbiting effects, emphasized by the reduced mass of the system, that indeed disappear in the $Ne^*(^3P) + NH_3$ CHEMI[37].

In summary, in the next sections the internally consistent formulation of the multidimensional Optical potential, that triggers CHEMI for three different systems, is used to cast light on fundamental details of their reaction stereo-dynamics, as those determining different value and energy dependence of total ionization cross sections. Moreover, the use of specific cuts of the potential, controlling the formation of the precursor (pre-reactive) state with particular energy and structure, is basic to emphasize the different role of two elementary reaction mechanisms with its dependence on collision energy, spin-orbit level of $Ne^*(^3P_{2,0})$ reagent, and orbital angular momentum (impact parameter) accompanying any two-body collision event. Such a difference, basic for the stereo-dynamics topology, casts light also on some limitations of the traditional capture models (for the reactivity at low temperature), that exclusively exploit the selectivity of the long-range forces combined with the effect of the centrifugal barrier.

## Results and discussion
### Observables and selectivity of the collision dynamics.
Penning ionization electron spectra (PIES), total ($\sigma$), and partial ($\sigma_p$) ionization cross sections, measured as a function of $E_{coll}$, and branching ratios (BRs) are available for a variety of CHEMI, investigated with MB technique. It has been demonstrated that PIES represents a sort of TS spectroscopy[31,32], while $\sigma$ and $\sigma_p$, with their dependence on $E_{coll}$, more directly probe the features of the interaction potential affecting the dynamics in the entrance channels.

As nuclear processes, also CHEMI are assumed to be driven by an Optical potential[30], whose real part $V$ controls the dynamical evolution of the reagents during each collision event, while the imaginary component $\Gamma$ defines the reactivity, that is the probability of passage from neutral entrance to ionic exit channels for each configuration of the collision complex. Recently, the Optical potentials were formulated, in an internally consistent way and in analytical form, for $Ne^*(^3P_J) + Ar$[25], $Ne^*(^3P_J) + N_2$[34], and $Ne^*(^3P) + NH_3$[7,35]. Their formulation, including angular and radial dependences for both real and imaginary components, permits to control the stereo-dynamics of such prototype CHEMI at increasing complexity of the M reagent, passing from an atom to a polyatomic molecule. More specifically, the proposed Optical potentials, reproducing most of experimental findings, obtained in our and in other laboratories, are here exploited to cast light on the basic points of general interest for the reaction dynamics stressed above.

For $Ne^*(^3P_J) + Ar$, $Ne^*(^3P_J) + N_2$ the adopted treatment[25,34] resolves the dependence of the reactivity on the spin-orbit levels J of the reagent $Ne^*$ that correlate with different symmetries of PS. For $Ne^*(^3P) + NH_3$[35] the lack of experimental information suggests to consider only the average effect on J. However, in the latter case the measure at a defined $E_{coll}$ of the PIES[38], with the shift of the main peaks position respect to that associated to pure photo-ionization spectra, provides a direct test of PS stability. Specifically, the analysis of PIES led also to separate reaction channels leading to $NH_3^+$ in its ground $X$ and in the first excited electronic state $A$, with the subsequent fragmentation in $NH_2^+ + H$.

Figure 1 reports $\sigma$, calculated with the proposed Optical potentials, as a function of $E_{coll}$ which extends from sub-thermal up to hyper-thermal conditions: The real part of interaction potentials used for cross-section $\sigma$ calculations are shown in Fig. 2.

In the case of ammonia, calculated $\sigma$ has been separated in partial $\sigma_p$ components, promoted by the most representative configurations of the PS opening the transition to the ionic product in the two different electronic states[7,34]. Actually, as detailed in ref. [8], the most quantitative treatment of CHEMI involving polar hydrogenated molecules, as ammonia, should include the average on the acceptance angular cone of each reaction channel. The adopted simplified treatment is useful to better emphasize the change of stereo-dynamics operative along the two different reaction channels.

For the three CHEMI, plotted results, with their different absolute value and dependence on $E_{coll}$, are consistent with experimental results of Perugia[8,39], Lausanne[37], Eindhoven[40], and Pittsburgh[41] experiments. In particular, for the comparison with Perugia experiments see Fig. 3 of ref. [8] and Fig. 4 of ref. [34], with Eindhoven experiments, see again Fig. 4 in ref. [34], with Pittsburgh experiments, see Fig. 4 in ref. [25], and, finally, with Lausanne experiments see comments to Fig. 5 in ref. [35]. Obtained results suggest that the stereo-dynamics in the three cases must be rather different, with a change also within the two channels effective for ammonia. Pronounced differences are exhibited also by CHEMI of Ar and $N_2$, although in the entrance channels the long-range attraction is practically the same[39]. Such differences arise from the change of the Optical potential at intermediate and short separation distances and, consequently, of structure and reactivity of TS.

Cross-section values, reported in Fig. 1, also emphasize the higher reactivity, predicted for $Ne^*(^3P_0)$ in collision with $N_2$ and

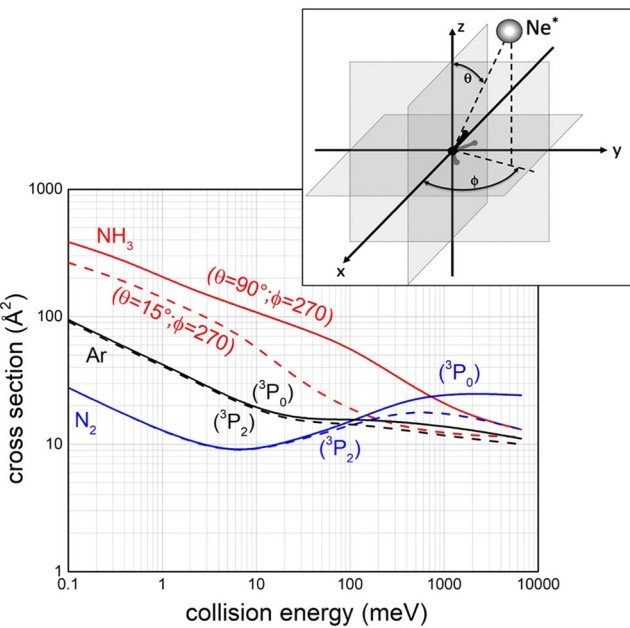

**Fig. 1 Total ionization cross sections for Ne\*($^3$P) + Ar, N$_2$, NH$_3$ CHEMI.** They are calculated as a function of the collision energy using the Optical potentials formulated in refs. [7,8,25,34,35]. For Ar and N$_2$ the different reactivity of Ne\*($^3$P$_2$) reagent (dashed lines) with respect to that of Ne\*($^3$P$_0$) (full lines), with its dependence on the collision energy, is also resolved, which appears to be consistent with the experimental findings[40]. For NH$_3$ the dependence of the reactivity on the precursor state geometry, opening the formation of NH$_3$$^+$ in its ground (full line) and in its first excited electronic state (dashed line), is shown. θ and φ are the polar angles as defined in the inset of the Figure.

Ar with respect to that of Ne\*($^3$P$_2$), emerging at high collision energies and confirmed by the Eindhoven experiments[40]. Here, we exploit specific cuts extracted from the multidimensional formulation of $V$ and $\Gamma$ components, to investigate new aspects of the stereo-dynamics topology more directly.

Figure 2 provides a comparison, on quantitative ground, of the interaction controlling the approach of reagents of investigated CHEMI. Selected $V$ components, that refer to specific cuts of the multidimensional potential energy surfaces, are useful to especially show the dependence on the intermolecular distance $R$ of the total potential $V_t$, defined as $V_t = V + V_c$, where the centrifugal potential $V_c$ is calculated using quantized $\ell$ values. Plotted results emphasize the capture efficiency of the reagents approaching at fixed $E_{coll}$ in the selected channels. From Fig. 2 it emerges that the centrifugal barrier strongly limits the value of the orbital angular momentum of the collision complex, defined by $\ell$, leading to the reaction. At the low $E_{coll}$ considered in such Figure, the centrifugal barrier becomes a sort of selector of orbital angular momentum effective for the reaction, permitting or hindering an approach of reagents sufficiently close to be effective for reactive events. This is clearer in the case of ammonia, where all reactive events, leading to the formation of NH$_3$$^+$(X), are controlled by low values of $\ell$ that determine distances $R_c$ very similar and confined at low values ($R_c$~2.4 Å). If $\ell$ gets over an upper limit, $R_c$ jumps at much higher values ($R_c > 10$ Å) where the reaction probability vanishes, being the reagents too distant, and the intermolecular forces null. In addition, the $R_c$ values ($R_c > 4$ Å) obtained for Ar and N$_2$ confirm the selective role of the centrifugal barrier and suggest that the chemical interaction components at such $R$ values play a marginal role, even at low $\ell$ values. For ammonia, the stereo-dynamics in the channel leading

to the formation of NH$_3$$^+$(A) shows a behavior intermediate between the cases analyzed above.

The semi-classical relation, $b \simeq \frac{\ell}{k}$, where $k$ is the wave number associated to the collision event, allows to better characterize important details concerning the dependence of $R_c$ on $b$ or on $\ell$. Such dependence arises from a critical balance between $V_t = V + V_c$ and $E_{coll}$.

Figure 3 quantifies the effects of the centrifugal component of the interaction, highlighted in Fig. 2, on the reaction dynamics of the investigated systems.

Specifically, Fig. 3 plots the $R_c$ dependence on the impact parameter $b$, evaluated, for the four cuts of $V$, at five $E_{coll}$ values that cover 4 orders of magnitude. At the lowest collision energies, plotted data confirm in all cases the selective role of the centrifugal barrier, which generates $R_c$ values confined in well-separated ranges, where the reaction probability is completely different. At the highest collision energies, a unique extended range of $R_c$ values becomes effective.

Figure 3 shows that CHEMI of ammonia, leading to the formation of the molecular ion in the ground electronic state NH$_3$$^+$(X), is exclusively controlled by short-range forces for $E_{coll} \leq 100$ meV. In particular, the reaction effectively occurs when $b$ assumes small and intermediate values and $R_c$ is confined at short range. Therefore, while the collision dynamics of reagents is fully controlled by the market capture from the strong attraction, determined by the combination of attractive electrostatic, induction, and dispersion components, the reactivity fully depends on the strength of chemical forces operative at short distances. An opposite behavior is exhibited by CHEMI of Ar and N$_2$ at very low $E_{coll}$, since their dynamics and reactivity are fully affected by the balance of weak forces operative at intermediate and long range, as further detailed in the next section. An intermediate behavior is confirmed for the reaction channel of ammonia leading to the NH$_3$$^+$(A) formation.

**Further selectivity.** Additional details on the topology of the reaction stereo-dynamics, are suggested by the Fig. 4, where, in the upper panel, are plotted the $\Gamma$ components, promoting DM and IM mechanisms, referred to the reaction channels investigated in this study (see Figs. 2 and 3). Such channels lead to the formation of product ions in the ground electronic state that for Ar$^+$($^2$P$_{3/2}$), an open shell atomic ion, is defined by $\Omega = 1/2$ ($\Omega$ is the quantum number representing the absolute projection of **J** vector along **R**).

The relative role of DM and IM mechanisms, defined as branching ratio (BR), is reported as a function of $R$ in the intermediate part of the same Figure. Reported BR quantifies the relative role of DM and IM in the vicinity of $R_c$ values that represent, as seen before, the distances most effective for CHEMI.

The Optical potentials formulation provides also the $C_x$ coefficient, which represents an important *marker-tracing* of the electronic evolution of the reagents as a function of their separation distance[24,25,34]. In particular, $C_x$ quantifies the $\Sigma$ molecular character degree assumed by PS formed via the collision of reagents. Note that a pure $\Sigma$ state is a true molecular state defined by the quantum number $\Lambda = 0$ (see also below). Therefore, the $C_x$ value indicates if PS is confined in the structure of two separated partners (quantified by the asymptotic 1/3 and 2/3 values), in that of a weakly interacting adduct (where $C_x$ starts to vary) or in that of a true molecule with a defined electronic symmetry (where $C_x$ assumes 1 or 0 as constant value), or in the transition between limit structures (where $C_x$ strongly varies). Such a detailed analysis has been carried out only for CHEMI of Ar and N$_2$, for which the different reactivity of Ne\* reagent in $^3$P$_0$ and $^3$P$_2$ fine levels has been resolved (see Fig. 1). The $C_x$

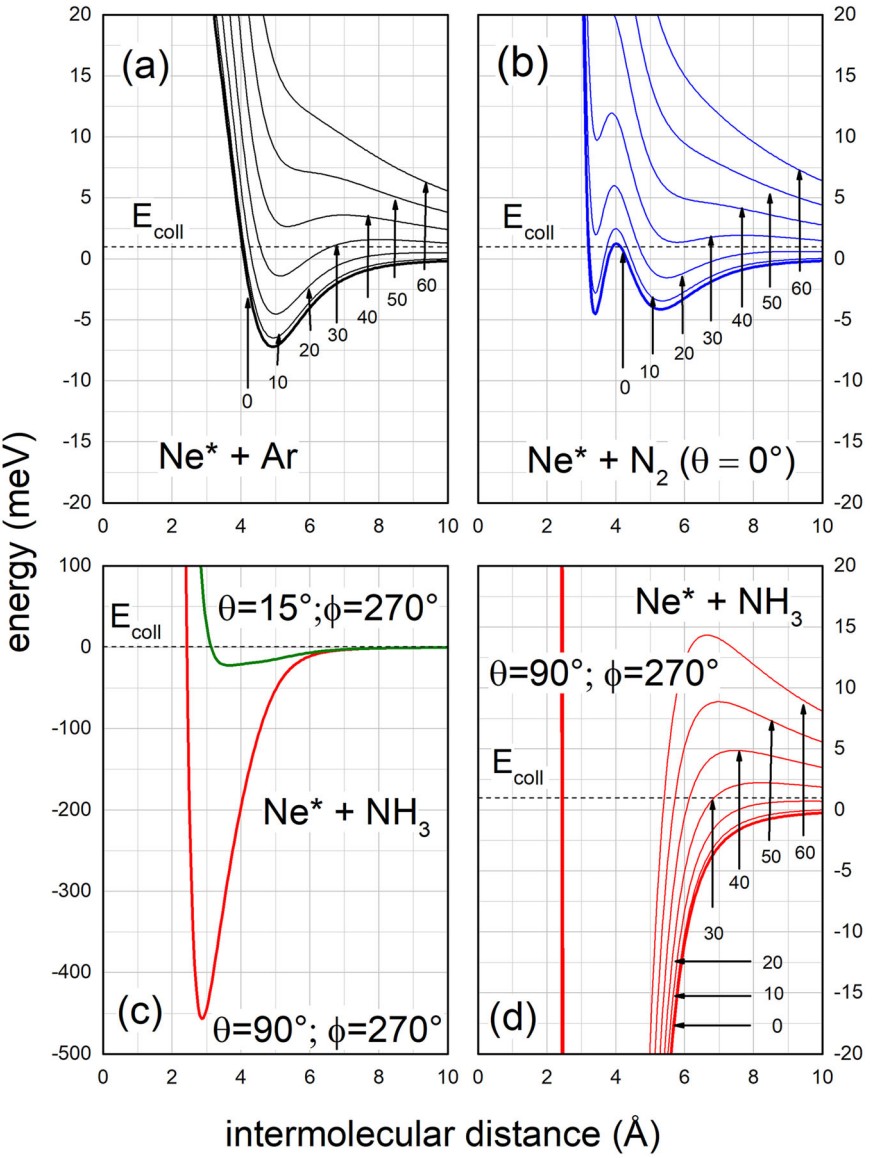

**Fig. 2 Interaction potentials controlling Ne\* + Ar, N$_2$, NH$_3$ investigated CHEMI.** Dependence on the intermolecular distance $R$ of the total potential energy $V_t$ given as sum of the real component $V$ and of the centrifugal contribution $V_C$, defined as $V_C = \frac{\hbar^2}{2\mu} \cdot \frac{\ell(\ell+1)}{R^2}$, where $\mu$ is the reduced mass of each system. θ and ϕ are the polar orientation angles of the molecule, as defined in Fig. 1 for NH$_3$[7,8,34,35]. The $E_{coll}$ value has been fixed to 1.0 meV and represented by a horizontal dashed line. For both Ne\*-Ar (**a**) and Ne\*-N$_2$ (**b**) the $V$ component is that asymptotically correlating with Ne($^3$P$_0$) fine level of the reagent[25,34], while for Ne\*-NH$_3$ (**c**, **d**), the plotted $V$ components account for the formation of the precursor state in two basic geometries, opening different CHEMI channels[35].

coefficient, plotted in the lower panel of Fig. 4, relates to the collinear configuration of Ne\* + N$_2$ reaction: it exhibits a behavior like that operative in Ne\* + Ar. The two different lines refer, respectively, to the |1/2,1/2> state of the internal ionic state of the adduct, that asymptotically correlates with Ne\*($^3$P$_0$) reagent, and to the |3/2,1/2> internal ionic state of the same adduct that asymptotically correlates with Ne\*($^3$P$_2$) reagent. The two complementary behaviors suggest that, while PS formed by the Ne\*($^3$P$_0$) reagent at short range becomes a linear tri-atomic molecule with a Σ symmetry ($C_x = 1$), the one formed by Ne\*($^3$P$_2$) becomes a molecule with a Π symmetry ($\Lambda = 1, C_x = 0$). This means that in the two different symmetries the half-filled orbital of the internal ionic core Ne$^+$ is aligned parallel and perpendicular to the separation distance, and this causes a reactivity more efficient in the first case when chemical forces, whose strength strongly varies with $R$ (depending on the

overlap integral of valence orbital), come into play. Also note that at intermediate and large separation distance the two symmetries are mixed by the spin-orbit coupling, associated to the open shell nature of interacting partners, leading to the formation of atomic $J$ fine levels.

Moreover, the bottom of Fig. 4 provides a cartoon of DM and IM promoting CHEMI of N$_2$, whose relative importance strongly varies with $R$. In particular, DM, triggered by chemical forces operative at short $R$, requires a proper symmetry of the reagents molecular orbital in order to favor the direct electron transfer. On the other hand, IM, stimulated by weak intermolecular forces of chemical and physical nature, dominates at larger $R$, where a mixing of atomic/molecular states of different symmetry, promoted by polarization, spin-orbit, and Coriolis couplings (the latter related to centrifugal effects accompanying scattering events) manifests. It has been demonstrated[24,25,34] that effective $\Gamma$

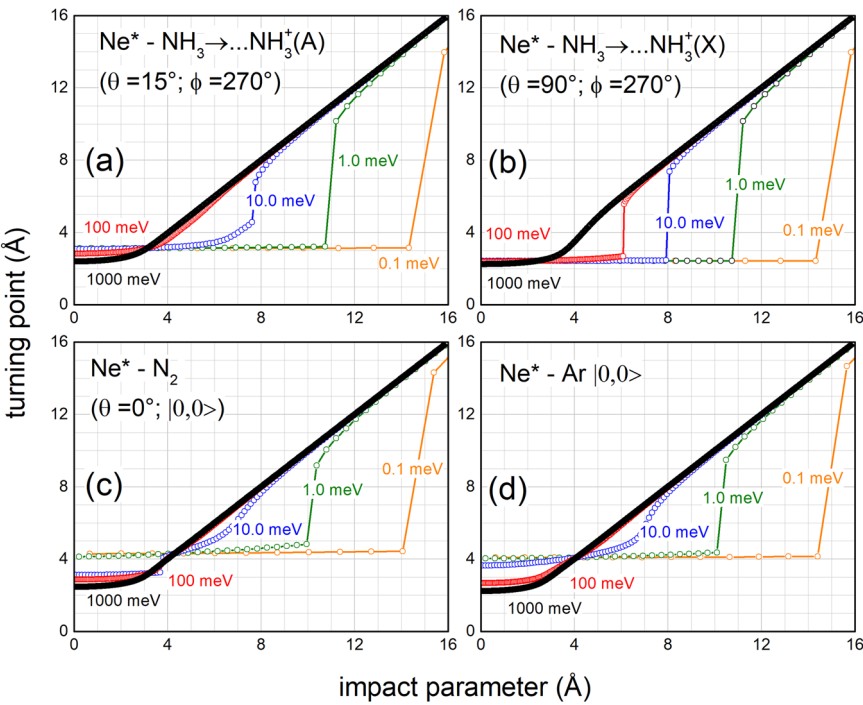

**Fig. 3 Effects of the centrifugal component of the interaction on Ne\* + Ar, N₂, NH₃ CHEMI.** The turning point ($R_c$) dependence on the impact parameter $b$, or, in quantum picture on $\ell$, evaluated for the four systems (Ne\*+NH₃ producing NH₃⁺(A) electronic excited ions (**a**), Ne\*+NH₃ producing NH₃⁺(X) electronic ground state ions (**b**), Ne\*+Ar (**c**) and Ne\*+N₂ (**d**) at five $E_{coll}$ values (see text). The ket indicates the selected atomic state of Ne\*(³P₀) reagent.

components are just a combination of $\Gamma_{DM}$ and $\Gamma_{IM}$ weighted on the $C_x$ marker-tracing coefficient plotted in the same Figure.

Additional information is suggested in Fig. 5, especially if analyzed together with the results in Figs. 2–4. In particular, the relative contribution of two reaction mechanisms can be obtained by combining the results of the turning points in Fig. 5 with the radial dependence DM and IM given in Fig. 4. Moreover, the strong attraction in the entrance channel and the centrifugal barrier, that acts as a selector of $b$ or values effective for the reaction (see Fig. 2), confirm that CHEMI of NH₃, leading to the formation of NH₃⁺ in the ground electronic state, is controlled exclusively by DM for $E_{coll}$ from 0.1 meV up to a value of 100 meV (see also Fig. 2, for $E_{coll}$ = 1 meV). For higher $E_{coll}$, IM plays some role only at large $\ell$ or $b$ values. Instead, for the same reaction, the channel leading to the formation of NH₃⁺ in the first excited electronic state is dominated by DM only for $E_{coll}$ up to few meV. For $E_{coll} \geq 10$ meV the reaction is determined by the balance of DM and IM and the DM/IM relative role increases with $E_{coll}$. Therefore, IM plays some role only in a limited window of intermediate $E_{coll}$ values. All these details are suggested by combining results in Figs. 3–5.

CHEMI of Ar behaves in an opposite way, that is the IM dominates at low collision energy, because of a less pronounced role of attraction components (see Fig. 1), here mostly controlled by dispersion forces. From Figs. 3 and 5 it appears that at $E_{coll} < 10$ meV the reaction exclusively occurs via IM, since the trapping effect of weak intermolecular forces is overcome at short range by the repulsion of size and of the centrifugal barrier. Consequently, the reagents remain always confined at large $R$. The transition to DM gradually occurs with the increase of $E_{coll}$ that permits a smooth passage to probing shorter $R_c$. This passage becomes evident for $E_{coll} \geq 100$ meV.

In the Ne\*(³P₀) + N₂ reaction the approach of reagents is affected by two potential wells of limited depth (4–5 meV, see Fig. 2): the first one manifests at about $R = 5.5$ Å, being mostly determined by the dispersion attraction; the second one occurs at about $R = 3.3$ Å and

arises from the combined effect of the floppy electronic cloud polarization of external atomic orbital $3s$ and of the interaction between the disclosed atomic internal ionic core and the permanent electric quadrupole of the anisotropic N₂ molecule. IM exclusively drives the reaction for $E_{coll}$ comparable/lower than 1 meV: because of the centrifugal barrier, the system probes exclusively the potential well at larger separation distance where IM dominates (see Figs. 3 and 5). DM promptly emerges at collision energy larger than 1 meV and becomes dominant already at $E_{coll}$ = 10 meV: here, the access to the second potential well, located at shorter $R$, is allowed (see Figs. 2 and 3). As for the other CHEMI, the role of IM becomes less relevant further increasing $E_{coll}$.

Finally, the analytical formulation of the anisotropic $V$ casts lights also on the natural molecular orientation (polarization) in the intermolecular electric field gradient with its effect on the reaction stereo-dynamics. The possibility of molecular polarization, during collisions driven by anisotropic intermolecular potentials, is usually evaluated exploiting the ratio between the average molecular rotation time, $\tau_{rot}$, and the collision time, $\tau_{coll}$ (see for instance ref. [37]). If $\tau_{rot} \ll \tau_{coll}$ the colliding system has time sufficient to assume the most stable configurations, while, when $\tau_{rot} \gg \tau_{coll}$, the molecule is seen under sudden conditions by the other colliding partner.

In general, $\tau_{coll}$ should be evaluated as

$$\tau_{coll} = \int_{R_{in}}^{R_c} \frac{dR}{v_r} \tag{2}$$

where

$$v_r = \left\{ \frac{2}{\mu} \left[ E \left( 1 - \frac{l(l+1)}{k^2 R^2} \right) - V(R) \right] \right\}^{1/2} \tag{3}$$

defines the radial velocity, $R_{in}$ represents an initial reference distance and $R_c$ is the closest approach distance (turning point). In the case of collisions driven by high anisotropic interactions, $v_r$ can undergo the levelling effect of $V(R)$ associated to the most

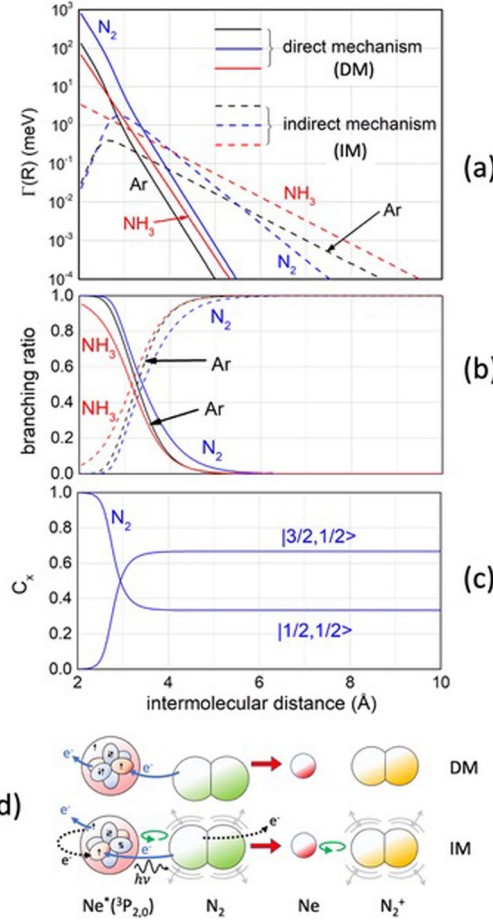

**Fig. 4 The role of DM and IM mechanisms on the topology of Ne\* + Ar, N₂, NH₃ stereo-dynamics. a** Comparison between the different dependence on intermolecular distance $R$ of $\Gamma$ components determining direct mechanism (DM) and indirect mechanism (IM) in the selected channels of CHEMI. The overall ionization width for each investigated system is approximately given by the sum of the two correlated $\Gamma$-functions plotted[35]. **b** Branching ratios, defined as $\Gamma_{DM}/(\Gamma_{DM} + \Gamma_{IM})$ and, $\Gamma_{IM}/(\Gamma_{DM} + \Gamma_{IM})$ are here plotted as a function of $R$. **c** The behavior of the $C_x$ marker tracing coefficient for Ne\*($^3P_{2,0}$) + N₂ CHEMI involving the precursor state in a collinear configuration. **d** A schematic cartoon illustrates the DM and IM mechanisms for the same reaction. DM, triggered by strong chemical forces, occurs with a direct electron transfer between valence orbital of reagents. IM, stimulated by anisotropic weak interactions as polarization, spin-orbit, and Coriolis contributions, involves a changing of the atomic/molecular alignment and a mixing of atomic levels and molecular states of different symmetry[23,29,41,47]. IM can promote, in addition to a stimulated electron transfer, also a virtual photon exchange between reagents giving to a photo-ionization.

attractive configurations. Under such conditions, $R_{in}$ can be taken as the distance where the interaction anisotropy is comparable with $E_{rot}$ and $\tau_{rot}$ can be substituted by $\tau_{pend}$ related to the formed libration-pendular states of the collision complex. Therefore, molecules rotationally relaxed, flying at low velocity and driven by strongly anisotropic interactions, in their collisions they are forced to assume the most stable configurations, undergoing the levelling effect of $V(R)$.

This treatment accounts for some findings of CHEMI of ammonia, where the BR between channels leading to NH₃⁺ in its ground $X$ and in first excited electronic state $A$ remains practically constant at a value 0.2-0.3 for $E_{coll}$ varying from 0.01up to 10 meV[37] and increases only for highest $E_{coll}$, that is when the

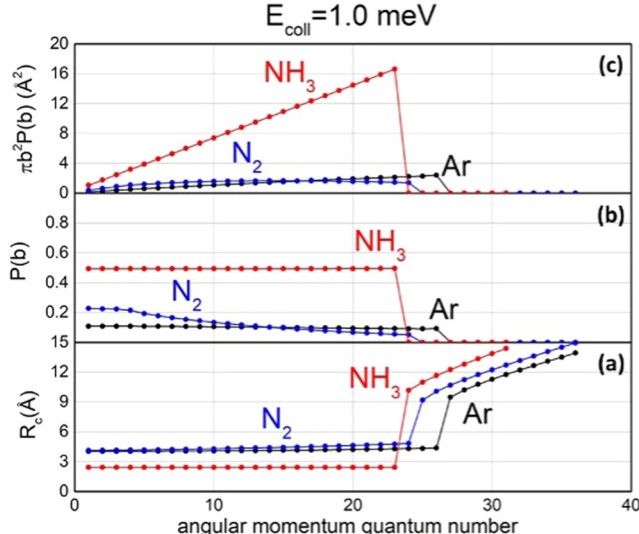

**Fig. 5 The role of the angular momentum quantum number on the selectivity of Ne\* + Ar, N₂, NH₃ reactivity.** Comparison between $R_c$ (**a**), the reaction probability $P(b)$ (**b**), and the ionization cross section contribution $\pi b^2 P(b)$ (**c**) evaluated as a function of $\ell/b$, at fixed $E_{coll} = 1.0$ meV. Note that for such $E_{coll}$ value the semiclassical relation $b \simeq \frac{\ell}{k}$ provides $b(\text{Å}) \simeq \frac{\ell}{2.097}$ for ammonia, $b(\text{Å}) \simeq \frac{\ell}{2.526}$ for Ar and $b(\text{Å}) \simeq \frac{\ell}{2.362}$ for N₂. For the reaction of ammonia, only the channel leading to the formation of NH₃⁺ in the ground state X is considered.

levelling effect of the $V$ attraction on $v_r$ vanishes[38]. Note that in the first case, molecules rotationally relaxed in seeded beams (with an average rotational period of some ps) have been used in low collision experiments[37], while in the second case[38] higher collision energies are probed with molecules in effusive beams, populating excited rotational levels.

In conclusion, for CHEMI this study emphasizes the important role of PS features, where the distance value of the closest approach $R_c$, with its dependence on the orbital angular momentum value, is crucial to define the relative role of DM and IM. This target has been achieved by exploiting a full, proper, and internally consistent formulation of the intermolecular interaction in some prototypical systems.

On the general ground, obtained results emphasize that at distances shorter than the centrifugal barrier location usually operate other selectivity, originated by the balance of short- and long-range intermolecular forces of different origin, not taken into account by traditional capture models, traditional Langevin[42] and Langevin modified models (see also ref. [37]). Some important results, concerning shortcomings of capture calculations by long-range dispersion forces have been recently reported for He\*($^3S,^1S$) + Li CHEMI[43]. The cases investigated in the present study are more complex, since simultaneously involving, as reagents, an open-shell P atom, with a high electron affinity ionic core, and molecular partners, with a permanent electric multipole. Related long-range forces arise from the balance between dispersion and several other anisotropic interaction components.

For CHEMI a key quantity is $R_c$ with its neighborhood, for other processes is $R_c$, and its comparison which crossing points, between potential energy surfaces having different symmetry and/or spin multiplicity, opening non-adiabatic transition towards specific exit channels[44]. All these details, combined with the role of natural molecular orientation originated by anisotropic intermolecular forces, are crucial to rationalize hot topics, as Arrhenius and anti-Arrhenius behavior of the chemical reactivity, especially affecting the cold chemistry[45].

Finally, a few years ago important stereo-dynamical effects were observed for $He^*(^3P_2) + H_2$ CHEMI under sub-thermal conditions[46]. Such effects have been ascribed to long- and short-range anisotropic interactions, controlled by the outer-occupied (strongly expanded) $2p$ orbital of $He^*$ and isotropic ground rotational state of para $H_2$ and by the two helicity states of ortho $H_2$ in its ground rotational state. The role of van der Waals and electrostatic effects (due to the quadrupole-quadrupole component) has been properly discussed. Present results cast light on the stereo-dynamics of $Ne^*(^3P_{2,0})$ + atom, molecule CHEMI where the internal ionic core $2p^5$ of $Ne^*$ is isoelectronic of F atom with an increased electron affinity, while the outer electron is confined in the symmetric 3s atomic orbital. As a consequence, while the long-range forces are here essentially isotropic, most part of the anisotropy arises at an intermediate and short separation distance, where the PS is formed and the orientation of ionic core with respect to the atomic-molecular partner plays a crucial role.

## Methods

**Experimental technique**. The experimental data discussed in the paper, have been measured by the MB technique in various laboratories but in all cases working in single collision conditions and at high resolution in angle and velocity[22,38,40,41] (see also Supplementary References). A scheme of our MB machine available in the Perugia laboratory is given in Supplementary Fig. 1. The main characteristics of the experimental device are mentioned in the Supplementary Methods: further details can be found in a previous paper[23] and in Supplementary References.

**Cross section calculations**. To assess all probed scattering properties, the semi-classical method, applicable until the de Broglie wave length of the system assumes values shorter or comparable to 1 Å and described elsewhere[24,25,30], has been used. The ionization cross sections are calculated using the Optical Model (see below) exploiting Supplementary Equations 1–5 discussed in the Supplementary Methods.

**Optical potential formulation**. The Optical Potential Model is defined in Supplementary Equation 1. Its real part is provided by a generalization of open-shell atom interactions discussed in the Supplementary Methods and expressed analytically by Supplementary Equations 6–9. The adopted potential formulation takes into account of a non-covalent component of the interaction as well as of an anisotropic term that includes also contributions of a chemical origin (see the case of $Ne^*$-$NH_3$ CHEMI of Fig. 2). In the case of the imaginary part of the Optical potential, the formulation has been discussed in previous papers[24,25,34,35] and same crucial details are emphasized in the present internally consistent analysis of the three different CHEMI dynamics (see, in particular, Fig. 4).

## Data availability

The authors declare that the data supporting the findings of this study are available within the paper and its Supplementary Information and from the corresponding author upon reasonable request.

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

## Acknowledgements

This work was supported and financed with the Fondo Ricerca di Base, 2019, dell'Università degli Studi di Perugia (Project Titled: Produzione di metano per reazione di idrogenazione di $CO_2$ con e senza catalisi in fase solida mediante l'uso di energie rinnovabili). Support from Italian MIUR and University of Perugia (Italy) is acknowledged within the program Dipartimenti di Eccellenza 2018-2022.

## Author contributions

S.F., F.V., and F.P. conceived and designed the study. S.F. performed experiments. S.F., F.V., and F.P. developed the proposed model with related calculations and analysed the experimental data. All authors participated in the writing and editing the paper.

## Competing interests

The authors declare no competing interests.
