## [Peer Review File · Communications Chemistry]

Reviewers' comments:

Reviewer #1 (Remarks to the Author):

In this work the authors detail the topology of the reaction stereo-dynamics in chemi-ionizations and ion chemistry. The work may be of interest to the wider scientific community (chemistry, physics, modeling of the various atmospheres and interstellar gas...). The paper is well written and clear and addresses specific gaps in the research area. However, in order to accept this paper for publication authors should make several improvements and clarifications.

Page 2, first sentence, typo: 13and references => 13 and references.

Next paragraph: X*species => X* species

Page 2, reaction $X^* + M \rightarrow \dots$ should be numbered.

Page 2, second sentence after reaction: TS => transition state (TS).
Acronyms/Abbreviations/Initialisms should be defined the first time they appear

Last sentence on page 2: X*can => X* can

Page 2, paragraph 2, second sentence: Please provide references to support those statements about each environment, i.e. interstellar medium, planetary atmospheres, flames and plasmas.

It would be good to label each panel in Figures 2, 3 and 5.

Can the authors further elaborate their statement "Figure 3 better quantifies some suggestions from Figure 2." page 10.

I would like the authors to explain further why CHEMI of Ar behaves in an opposite way. Page 15 first sentence.

Page 16: (See below) => (see below)

Page 17: Eq. should be numbered. Or if it is not important just put direct in the text.

References are appropriate, but due to their number they should be checked carefully. In ref. [45] standard abbreviation (ISO 4) for journal Astronomy & Astrophysics is Astron. Astrophys. not A&A.

If you didn't provide doi numbers for the 46 references, then you don't need then for the 47th reference either. Missing vol in Ref [47]. 7(11)

Reviewer #2 (Remarks to the Author):

Chemi-ionization processes, in which an atom or molecule is ionized by a highly excited atom or molecule, are, for instance, relevant for the chemistry of flames and planetary atmospheres. Chemi-ionization has already been studied for many years, but recent experimental work has revived interest in these processes. Falcinelli et al. report on the results of semi-classical calculations of chemi-ionization reactions ($Ne^* + NH_3$, $Ne^* + Ar$, $Ne^* + N_2$) which have also been subject to experimental work. The three systems are of particular interest, since they have different short- and long-range

behaviors, so that the two mechanisms that govern chemi-ionization (electron exchange and radiative transfer) come into play in different regimes.

The results are novel and certainly interesting for the specialist audience. However, in its current state, the article has several shortcomings and is very hard to follow. Therefore, I recommend that the article is substantially revised before publication.

The following points must be addressed:

Owing to the influence of the charge, ion-molecule systems have very different interaction potentials compared to atom-molecule systems. The given link in between chemi-ionization (involving only neutral species) and ion-molecule chemistry is therefore not obvious to me, and appears too ambitious. Either the authors provide sound evidence in their article of why their theory approach can also be applied to ion-molecule chemistry, or these statements have to be omitted. Since the article only deals with calculations of chemi-ionization reactions, the title is misleading and should be changed.

The abstract and the first part of the introduction read like a review article, and, until the last paragraph of page 3, I could not tell that this article would be dealing with the results of semi-classical calculations of chemi-ionization reactions. The abstract must be re-written.

The article is full of information that is only accessible to experts in collision dynamics. For instance, the authors provide no technical details about how the semi-classical calculations are done. I foresee that the non-expert reader will have a hard time to understand this work without having to consult the literature.

It is generally known that the centrifugal barrier limits chemical reactivity at low collision energies. I do not see the novelty of the results reported in Fig. 2 and in the text associated with it.

The authors state that "Their formulation [...] permits to control such prototype CHEMI at increasing complexity of the M reagent." It is unclear how this control should be achieved.

The authors state that the cross sections obtained from their calculations as well as their energy dependence are consistent with previous experimental results (from Perugia, Lausanne, Eindhoven). However, the results from previous work are not shown (e.g. in a table or in one of the figures given in the article). Hence the reader would have to consult the literature for this. The authors should add the experimental results for comparison.

The description of the timescales for collision and rotation is quite lengthy given that it is only used for a five-line-short comparison with experimental results. It is also very difficult to link the descriptions for the collision and rotation timescales with the explanation for the energy dependence of the $\text{Ne}^+ + \text{NH}_3$ reaction given thereafter.

I have counted 14 self-citations in this article. Considering that chemi-ionization has been subject to more than 50 years of research, I doubt that all of these are strictly required. Likewise, some references are missing. Examples:

a) In the introduction, the two mechanisms are introduced. The mechanisms have been known for many years, and reference should be given to this previous work, see, e.g. the work by Hotop and Niehaus (Z. Physik 228, 68-88, 1969) and references in Siska (Rev. Mod. Phys. 65, 337-412, 1993).

b) Recent work by Dulitz et al. (Phys. Rev. A 102, 022818, 2020) also reports on the shortcomings of capture calculations for the description of chemi-ionization.

c) "This is confirmed by the observation of orbiting resonances in $\text{He}^* + \text{NH}_3$ ". There is earlier work on orbiting resonances in chemi-ionization (Henson et al., Science 338, 6104, 234-238, 2012) which should be cited.

I strongly recommend having the article proofread and edited by a native English speaker to improve readability.

Minor points:

In Fig. 4, it would also be nice to plot the overall ionization widths.

Cartoon in Fig. 4: It would be easier to understand if the chemical formulas were below the graphics.

Reviewer #3 (Remarks to the Author):

In this manuscript, the authors study the stereodynamics of chemi-ionization reactions between atomic Ne (in the electronic excited 3P state) and Ar, N₂ and NH₃.

Based on the intermolecular interaction potentials obtained previously, here the authors carry out molecular dynamics simulations that allowed them to conclude that the outcome of the collision is determined by the properties of the precursor state (the intermolecular complex) which acts as a transition state of this process. In particular, they suggest that the centrifugal barrier determines the relative weight of the direct and indirect mechanisms.

In my opinion, the manuscript does not represent a clear advance, and it is not likely to influence thinking in the field so I cannot recommend it for its publication in Communications Chemistry:

** It seems that the most important part of the results is the interaction potentials that were calculated previously. The results of the dynamic simulations are shown in Figure 5 (those in Fig.1 are restricted to particular initial geometries of NH₃-Ne), but it is not possible to extract from them the contributions of the direct or indirect mechanism.

** The methodology used is not sufficiently explained in the manuscript. In particular, I could not find enough details about how the dynamic simulations were carried out.

** The authors state that around the closest approach distance the system spends most part of its collision time. However, it seems that trapping in the potential well may play a very important role, in particular for the collisions with NH₃. In fact, according to Fig.5, most of the scattering comes from not significantly large values of L. On the same topic, resonances are probably occurring at those energies, leading to the trapping of the intermolecular complex.

** I found that some parts of the manuscript are confusing. For example:

** it is not clear which are the components of the imaginary part of the potential that promote the direct and indirect mechanism, and how they were calculated.

** It is not clear why the authors emphasize collisions for Ne + N₂ at $\theta=0$ and not average over all the possible geometries. The same applies for the collisions with NH₃, where it is also not clear why those two initial geometries were highlighted.

Responses to the Reviewer #1 comments:

Reviewer #1 – general comments: *“In this work the authors detail the topology of the reaction stereo-dynamics in chemi-ionizations and ion chemistry. The work may be of interest to the wider scientific community (chemistry, physics, modeling of the various atmospheres and interstellar gas...). The paper is well written and clear and addresses specific gaps in the research area. However, in order to accept this paper for publication authors should make several improvements and clarifications.”*

Authors reply and made modifications: We thank the reviewer #1 for his positive comments and related useful suggestions which have been fully addressed by us (see below)

Reviewer #1 - addressed point 1: “Page 2, first sentence, typo: 13and references => 13 and references.”

Authors reply and made modifications: We thank the reviewer and we have corrected the first sentence at page 2.

Reviewer #1 – addressed point 2: “Next paragraph: X^* species => X^* species.”

Authors reply and made modifications: same as above.

Reviewer #1 – addressed point 3: “Page 2, reaction $X^* + M \rightarrow \dots$ should be numbered.”

Authors reply and made modifications: We thank the reviewer and the reaction at page 3 of the revised manuscript version has been numbered as equation (1).

Reviewer #1 – addressed point 4: “Page 2, second sentence after reaction: TS => transition state (TS). Acronyms/Abbreviations /Initialisms should be defined the first time they appear.”

Authors reply and made modifications: We thank the reviewer for his comment, and we defined the acronym TS just after the equation (1) which is the right place in the text.

Reviewer #1 – addressed point 5: “Last sentence on page 2: X^* can => X^* can.”

Authors reply and made modifications: same as points 1 and 2 above.

Reviewer #1 – addressed point 6: “Page 2, paragraph 2, second sentence: Please provide references to support those statements about each environment, i.e. interstellar medium, planetary atmospheres, flames and plasmas.”

Authors reply and made modifications: We agree with the reviewer and we added the proper references modifying the related sentence at page 4, line 7 from the top of the revised manuscript version as it follows: “They happen both at very low temperature T (under conditions of the cold chemistry)^{18,21}, as in interstellar medium, at intermediate T values, as in planetary atmospheres,²⁵ and at high T, as in flames and plasmas (see Refs. 24,25 and references therein).”.

Reviewer #1 – addressed point 7: “It would be good to label each panel in Figures 2, 3 and 5.”

Authors reply and made modifications: We agree with the reviewer and we modified Figs. 2, 3 and 5 (and related captions), labelling each panel as (a), (b), (c) and (d).

Reviewer #1 – addressed point 8: “Can the authors further elaborate their statement “Figure 3 better quantifies some suggestions from Figure 2. “page 10.”

Authors reply and made modifications: We thank the reviewer and, in order to be more clear, we replaced the old sentence “Figure 3 better quantifies some suggestions from Figure 2” with the following new sentence (see the revised manuscript at page 12 just before Fig. 3): “The Fig. 3 quantify the effects of the centrifugal component of the interaction, highlighted in Fig. 2, on the reaction dynamics of the investigated systems.”.

Reviewer #1 – addressed point 9: “I would like the authors to explain further why CHEMI of Ar behaves in an opposite way. Page 15 first sentence.”

Authors reply and made modifications: We thank the reviewer and we modified the related sentence (see page 17, first line on the top of the revised manuscript) as it

follows: “CHEMI of Ar behaves in an opposite way, **that is the IM dominates at low collision energy**, because of a less pronounced role of attraction components (see Figure 1), here mostly controlled by dispersion forces.”

Reviewer #1 – addressed point 10: “Page 16: (See below) => (see below).” **Authors reply and made modifications:** We have replaced the word “See” with “see” throughout the text.

Reviewer #1 – addressed point 11: “Page 17: Eq. should be numbered. Or if it is not important just put direct in the text.”

Authors reply and made modifications: We agree with the reviewer for his comment, and the reactions at the end of page 18 of the revised manuscript have been numbered as equations (2) and (3).

Reviewer #1 – addressed point 12: “References are appropriate, but due to their number they should be checked carefully. In ref. [45] standard abbreviation (ISO 4) for journal *Astronomy & Astrophysics* is *Astron. Astrophys.* not *A&A.*”

Authors reply and made modifications: We apologize for that error and have corrected the old ref. 45 (now ref. 46 in the revised manuscript) following the reviewer's suggestion.

Reviewer #1 – addressed point 13: “If you didn't provide doi numbers for the 46 references, then you don't need then for the 47th reference either. Missing vol in Ref [47]. 7(11).”

Authors reply and made modifications: We thank the reviewer and have included the ref. 47 correctly as it follows: 47. Shagam, Y.; Klein, A.; Skomorowski, W.; Yun, R.; Averbukh, V.; Koch, C. P.; Narevicius E. Molecular hydrogen interacts more strongly when rotationally excited at low temperatures leading to faster reactions. *Nat. Chem.* **2015**, *7*, 921-926.

Responses to the Reviewer #2 comments:

Reviewer #2 – general comment: “Chemi-ionization processes, in which an atom or molecule is ionized by a highly excited atom or molecule, are, for instance, relevant for the chemistry of flames and planetary atmospheres. Chemiiionization has already been studied for many years, but recent experimental work has revived interest in these processes. Falcinelli et al. report on the results of semi-classical calculations of chemi-ionization reactions ($Ne^* + NH_3$, $Ne^* + Ar$, $Ne^* + N_2$) which have also been subject to experimental work. The three systems are of particular interest, since they have different short- and long-range behaviors, so that the two mechanisms that govern chemi-ionization (electron exchange and radiative transfer) come into play in different regimes. The results are novel and certainly interesting for the specialist audience. However, in its current state, the article has several shortcomings and is very hard to follow. Therefore, I recommend that the article is substantially revised before publication. The following points must be addressed:...”

Authors reply and made modifications: We thank the reviewer #2 for his positive and encouraging comments and criticisms. We revised our manuscript addressing all his comments and suggestions as it follows.

Reviewer #2 – addressed point 1: “Owing to the influence of the charge, ion-molecule systems have very different interaction potentials compared to atom-molecule systems. The given link in between chemi-ionization (involving only neutral species) and ion-molecule chemistry is therefore not obvious to me, and appears too ambitious. Either the authors provide sound evidence in their article of why their theory approach can also be applied to ion-molecule chemistry, or these statements have to be omitted. Since the article only deals with calculations of chemi-ionization reactions, the title is misleading and should be changed.”

Authors reply and made modifications: We thank the reviewer for his criticism that allows us to avoid possible misunderstandings and to better clarify the link between chemi-ionization and ion-molecule reactions in our manuscript. For the first purpose, we changed the title avoiding to mention the ion chemistry, as it follows: “**The topology of the reaction stereo-dynamics in chemi-ionizations**”. In order to satisfy the second purpose, we added the following sentence at page 3, line 6 from the bottom of the revised manuscript: “**It is relevant to note that because of an external electronic cloud polarization, the PS $[X\cdots M]^*$ formed at short separation distances tends to assume the configuration of an internal ionic core surrounded by a Rydberg electron. Therefore, DM can be considered as a particular case of ion chemistry, triggered by non-resonant charge transfer coupling entrance and exit channels.²⁵**”.

Reviewer #2 – addressed point 2: “The abstract and the first part of the introduction read like a review article, and, until the last paragraph of page 3, I could not tell that this article would be dealing with the results of semiclassical calculations of chemi-ionization reactions. The abstract must be re-written.”

Authors reply and made modifications: We agree with the reviewer and the abstract has been rewritten as it follows: “**In this paper, an accurate internally consistent formulation of the optical interaction potentials, obtained by the combined analysis of scattering and spectroscopic experimental findings, casts light on structure, energy and angular momentum couplings of the precursor (pre-reactive) state controlling the stereo-dynamics of prototypical chemi-ionization reactions.**

Attention is focused on the orbital angular momentum selectivity, affecting the fate of each reactive event at any collision energy. The closest approach (turning point) of reagents, is found to control the relative weight of two different reaction mechanisms that exhibit a completely different dependence on the reagents separation distance: i) **An direct mechanism stimulated by exchange chemical forces mainly acting at short separation distances and high collision energy; ii) An indirect mechanism, caused by the combination of weak chemical and physical forces dominant at larger distances, mainly probed at low collision energy, that can be triggered by a “virtual” photon exchange between reagents.**”.

Furthermore, also the Introduction section has been reorganized in order to highlight novelty and importance of the paper, addressing comments and criticisms from either reviewers#2 and #3. For such a purpose, the following sentences have been added:

- at page 2, line 6 from the bottom of the revised manuscript: “**In particular, the focus is on barrier-less chemi-ionization reactions in order to provide unique-direct information on basic quantities determining the topology of their stereo-dynamics. The precursor state (PS), formed by collisions of reagents, plays here an important role being coincident with the reaction transition state. All the features of such a state, as well as its structure and stability, are directly or**

indirectly controlled by intermolecular forces operative in each collision complex.”;

- at the end of page 3, line 6 from the bottom of the revised manuscript: “It is relevant to note that because of an external electronic cloud polarization, the PS $[X \cdots M]^*$ formed at short separation distances tends to assume the configuration of an internal ionic core surrounded by a Rydberg electron. Therefore, DM can be considered as a particular case of ion chemistry, triggered by non-resonant charge transfer coupling entrance and exit channels.²⁵”;
- at page 4, line 5 from the bottom of the revised manuscript: “The present study exploits an accurate formulation of the intermolecular interaction potentials, previously obtained^{23-25,33} and suggested by the complex phenomenology of open shell “P” atoms, investigated by advanced experimental and theoretical methods, by the behaviour of ion-neutral systems, coupled by charge transfer, and by the spectroscopic properties of excimers. The obtained formulation has been tested on experimental findings of CHEMI, investigated in our and other laboratories by coupling scattering and spectroscopic techniques.^{7,8,23-25,34,35} In order to emphasize innovative aspects of the reaction stereo-dynamics, we have found convenient to refer to particular geometries of PS (or pre-reactive state), that open specific reaction channels. However, in the analysis and tests on the experimental findings, the full space of the relative geometries of reagents has been considered. In addition, in the case of NH_3 CHEMI the two considered geometries are the most relevant ones promoting, within selected angular cones, the formation of NH_3^+ ionic product in the ground (X) and in the first excited (A) electronic states.^{7,8,35} The acceptance of angular cones is discussed in detail in Ref. 8. Therefore, the selection of individual reaction channels, triggered by specific cuts of the multi-dimensional interaction potential has been useful to properly address...”;
- at page 6, line 4 from the top of the revised manuscript: “Thanks to the above, we are able to perform a first original attempt to point out how the chemical reactivity depends on the angular momentum quantum number of the intermediate collision complex (precursor or pre-reactive state) leading to the reaction. In particular, we are able to clarify the way in which the centrifugal barrier of the colliding reagents strongly acts as a sort of “selector” of the orbital quantum number effective for the reaction. In our knowledge, this is the first attempt to depict the topology of the stereo-dynamics of a chemical reaction by a state-to-state treatment.”;
- at page 6, line 12 from the bottom of the revised manuscript: “Furthermore, obtained results suggest some limitations of the so called “capture models”, usually exploited to describe many other barrier-less processes, like ion–molecule reactions, occurring in interstellar medium, planetary atmospheres and plasmas.”.

Reviewer #2 – addressed point 3: *“The article is full of information that is only accessible to experts in collision dynamics. For instance, the authors provide no technical details about how the semi-classical calculations are done. I foresee that the non-expert reader will have a hard time to understand this work without having to consult the literature.”*

Authors reply and made modifications: In order to address this comment of the reviewer, we added:

- i) a new Methods section at the end of page 20 of the revised manuscript, as requested by the Journal, whose text is the following:

“Methods

Experimental technique

The experimental data discussed in the paper, have been measured by the MB technique in various laboratories but in all cases working in single collision conditions and at high resolution in angle and velocity^{22,38,40,41}. A scheme of our MB machine available in the Perugia laboratory is given in Supplementary Fig. 1. The main characteristics of the experimental device are mentioned in the Supplementary Methods: further details can be found in a previous paper²³.

Cross section calculations

To assess all probed scattering properties, the semiclassical method, applicable until the de Broglie wave length of the system assumes values shorter or comparable to 1 Å and described elsewhere^{24,25,30}, has been used. The ionization cross sections are calculated using the Optical Model (see below) exploiting Supplementary Equations 15 discussed in in the Supplementary Methods.

Optical potential formulation

The Optical Potential Model is defined in the Supplementary Equation 1. Its real part is provided by a generalization of open shell atom interactions discussed in the Supplementary Methods and expressed analytically by the Supplementary Equations 69. The adopted potential formulation takes into account of a non-covalent component of the interaction as well as of an anisotropic term that includes also contributions of a “chemical” origin (see the case of Ne* -NH₃ CHEMI of Fig. 2). In the case of the imaginary part of the Optical potential, the formulation has been discussed in previous papers^{24,25,34,35} and same crucial details are emphasized in the present internally consistent analysis of the three different CHEMI dynamics (see, in particular, Fig.4).

Data availability

The authors declare that [the/all other] data supporting the findings of this study are available within the paper [and its supplementary information files]. The data that support the findings of this study are available from the corresponding author upon reasonable request.”;

- ii) a related link to a Supplementary Information (SI) document to the revised manuscript, by the following sentence at page 6, line 5 from the bottom: “... whose general aspects are given in the Supplementary Methods section of SI.”.

The new text of the added linked SI is the following:

“Supplementary Methods

Additional details on the Optical Potential formulation

The theoretical treatment, presented in this paper and stimulated by the PIES and mass spectrometric measurements in our laboratory, is able to rationalize in a unifying picture all the observables quoted above, probing different features of the so called Optical Potential Model:¹⁻³

$$= -2\Gamma \quad (1)$$

Whose, the real part, $V_i(R)$, controls the dynamical evolution of the reagents during each collision event, while the imaginary component, $\Gamma(R)$, defines the reactivity, that is the probability of passage from neutral entrance to ionic exit channels for each configuration of the collision complex. This is quantified by the lifetime of the autoionizing system at the separation distance R expressed by the eq. (2):

$$\Gamma(R) = \frac{\hbar}{\tau(R)} \quad (2)$$

The ionization probability at any distance R is given by the following eq. (3):

$$P(R) = \exp\left[-\frac{\Gamma(R)}{\hbar \sqrt{1 - \frac{E}{E_2}}}\right] \quad (3)$$

where g represents the relative asymptotic velocity, E the collision energy and b the impact parameter.

During a complete collision, the probability that the system survives (i.e. does not give rise to ionization) from infinite distance to the collision turning point R_c is:

$$P_{\infty}(R_c) = \exp\left[-\int_{R_c}^{\infty} \Gamma(R) dR\right] = \exp\left[-\int_{R_c}^{\infty} \frac{\Gamma(R)}{\hbar \sqrt{1 - \frac{E}{E_2}}} dR\right] \quad (4)$$

Finally, the total ionization cross section is given by the eq. (5) below.

$$\sigma_{\text{ion}} = 2 \int_0^{\infty} P_{\infty}(R_c) b db = 2 \int_0^{\infty} \left[\exp\left[-\int_{R_c}^{\infty} \frac{\Gamma(R)}{\hbar \sqrt{1 - \frac{E}{E_2}}} dR\right] \right] b db \quad (5)$$

The adopted methodology is able to reproduce all experimental data available for the $\text{Ne}^+ - \text{Ar}$, N_2 and NH_3 systems obtained from our and other laboratories, including total and partial ionization cross sections and branching ratios between selected channels. Therefore, the proposed theoretical approach, which also includes within the same framework exchange and radiative mechanisms proposed in the past³, is general and can be used to describe in a state-to-state condition the reactivity of all chemi-ionization reactions, including those involving molecules.^{4,5}

For the real component of the Optical Potential of eq. (1), the adopted formulation providing for the entrance channels the dependence of the isotropic interaction on the reagent separation distance R , leads to this expression:⁶

$$V_i(R) = V_{\text{ILJ}}(R) + (1 - S(R)) V_{\text{ion}}(R) \quad (6)$$

where V_{ILJ} and V_{ion} are represented by the Improved Lennard Jones (ILJ) function, whose general form is:⁶

$$V_{\text{ILJ}}(R) = \left[\frac{\epsilon}{\left(\frac{R}{r_0}\right)^{12}} - \frac{1}{\left(\frac{R}{r_0}\right)^6} \right] S(R) \quad (8)$$

Here ϵ is the potential well depth and r_0 is its location, while n defines the hardness of the repulsive wall and the radial modulation of the attraction. The switching function $S(R)$, that accounts for the transition from the neutral-neutral to the ion-neutral representation of the interaction, as previously⁵ it has been defined as

$$S(R) = 1 + \left(\frac{R - d}{d}\right)^{-1} \quad (9)$$

Here d is the distance where the two combined limiting potential forms have the same weight, while d describes how fast the transition occurs.

The R dependence of the isotropic component of the interaction in the exit channel is represented again by an ILJ function. Basic aspects of the potential formulation for atom-atom reactions are discussed in a previous article⁷ where it is shown that the

parameter values were obtained from a semi-empirical method founded on the ample phenomenology of the interactions of open-shell “P” atoms (particularly halogen atoms). They have been investigated in detail with scattering experiments, performed with state selected atomic beams and analysed with a proper theoretical treatment.⁸⁻¹⁰ Details on the extension of the formulation to atom-molecule reactions, including the angular dependence of both terms of the optical potential W , are given in ref. 7 and in refs. 7,8,34,35 of the main text.

Additional Details on the experimental determinations

The experiments, performed under single collision condition with the molecular beam (MB) technique, allowed the measure of total and partial ionization cross sections¹¹⁻¹⁵, branching ratios^{16,17} and Penning Ionization Electron Spectra (PIES)^{15,18-21}

A scheme of the MB machine operating in our laboratory is shown in Supplementary Fig. 1. A primary beam of $\text{Ne}^*(^3\text{P}_1)$, with $J=2,0$ atoms, emerging from an electron bombardment effusive or supersonic seeded source, crosses at right angles the secondary beam of target particles (Ar, N_2 or NH_3). PIESs have been measured exploiting a hemispherical electron energy analyzer located above the beam crossing volume, while total, partial cross sections and branching ratios have been determined by mass spectrometry using a quadrupole mas filter placed below the beam scattering center. It consists of three vacuum chambers: the first one contains the rare gas beam source, while in the second chamber the rare gas atoms are electronically excited and pulsed by a slotted disk; in the third chamber the metastable atoms cross the target molecules of a secondary effusive beam. In this chamber the metastable atoms are monitored, while product ions and emitted electrons are detected, after mass analysis, for the ions, and energy selection, for the electrons. The neon beam can be produced by two sources that can be used alternately. The first one is a standard effusive source at room temperature while the second one is a supersonic device that can be heated to different temperatures. In both cases the metastable atoms are produced by electron bombardment at about 150 eV, that is expected to yield $\text{Ne}(^3\text{P}_2)$ and $\text{Ne}(^3\text{P}_0)$ in a population close to the statistical 5:1 ratio.⁴ The metastable atom velocity can be analyzed by a time-of-flight (TOF) technique. The resolution of our electron spectrometer is of about 45 meV at a transmission energy of 3 eV, as determined by measuring the photoelectron spectra of Ar, O_2 , and N_2 by He(I) radiation with the procedure described elsewhere.^{15,20} Spurious effects due to the geomagnetic field have been reduced to ≤ 20 mG by a μ -metal shielding.

Supplementary Fig. 1

The schematic view of the apparatus used for Ne^* -Ar, N_2 and NH_3 chemi-ionization studies located at University of Perugia.

Supplementary References

- [1] P. E. Siska, *Rev. Mod. Phys.* **1993**, *65*, 337-412.
- [2] B. Brunetti and F. Vecchiocattivi, *Current Topic on Ion Chemistry and Physics*, Ng, C. Y., Baer, T., Powis I., Eds.; John Wiley & Sons Ltd: New York, 1993; pp 359-445.
- [3] W. H. Miller and H.; Morgner, *J. Chem. Phys.* **1977**, *67*, 4923-4930.
- [4] S. Falcinelli, F. Vecchiocattivi, F. Pirani, *Phys. Rev. Lett.* **2018**, *121*, 163403.
- [5] B. G. Brunetti, P. Candori, S. Falcinelli, F. Pirani, F. Vecchiocattivi, *J. Chem. Phys.* **2013**, *139*, 164305.
- [0] F. Pirani, S. Brizi, L. F. Roncaratti, P. Casavecchia, D. Cappelletti, F. Vecchiocattivi, *Phys. Chem. Chem. Phys.* **2008**, *10*, 5489-5503.
- [6] S. Falcinelli, F. Vecchiocattivi, F. Pirani, *Commun. Chem.* **2020**, *3(1)*, 64.
- [7] V. Aquilanti, R. Candori, F. Pirani, *J. Chem. Phys.* **1988**, *89*, 6157-6164.
- [8] E. E. Nikitin and R. N. Zare, *Mol. Phys.* **1994**, *82*, 85-100.
- [1] F. Pirani, G. S. Maciel, D. Cappelletti, V. Aquilanti, *Int. Rev. Phys. Chem.* **2006**, *25*, 165-199.
- [2] H. Hotop, A. Niehaus, *Chem. Phys. Lett.* **1971**, *8*, 497-500.
- [3] V. Hoffmann, H. Morgner, *J. Phys. B: Atom. Molec. Phys.* **1979**, *12*, 2857-2874.
- [4] K. Ohno, H. Mutoh, Y. Harada, *J. Am. Chem. Soc.* **1983**, *105*, 4555-4561.
- [5] K. Ohno, *Bull. Chem. Soc. Japan* **2004**, *77*, 887-908.
- [6] B. G. Brunetti, P. Candori, D. Cappelletti, S. Falcinelli, F. Pirani, D. Stranges, F. Vecchiocattivi, *Chem. Phys. Lett.* **2012**, *539-540*, 19-23.

- [7] S. D. S. Gordon, J. Zou, S. Tanteri, J. Jankunas, A. Osterwalder, *Phys. Rev. Lett.* **2017**, *119*, 053001.
- [8] S. D. S. Gordon, J. J. Omiste, J. Zou, S. Tanteri, P. Brumer, A. Osterwalder, *Nat. Chem.* **2018**, *10*, 1190-1195.
- [9] K. Ohno, *Bull. Chem. Soc. Japan* **2004**, *77*, 887-908.
- [10] S. D. S. Gordon, J. Zou, S. Tanteri, J. Jankunas, A. Osterwalder, *Phys. Rev. Lett.* **2017**, *119*, 053001.
- [11] H. Hotop, *J. Electron. Spectrosc. Relat. Phenom.* **1981**, *23*, 347-365.
- [12] B. A. Jacobs, W. A. Rice, P. E. Siska, *J. Chem. Phys.* **2003**, *118*, 3124-3130.

Reviewer #2 – addressed point 4: *“It is generally known that the centrifugal barrier limits chemical reactivity at low collision energies. I do not see the novelty of the results reported in Fig. 2 and in the text associated with it.”*

Authors reply and made modifications: We agree with the reviewer but Fig.2 allows us to quantitatively compare the systems we have chosen to study. In fact, they involve very different PESs and this helps us to highlight the effect played by the centrifugal barrier when dealing with interaction potentials having completely different characteristics. To clarify this point, we modified the sentence at page 10, line 3 from the top of the revised manuscript as it follows: *“The Fig. 2 provides a comparison, on quantitative ground, of the interaction controlling the approach of reagents of investigated CHEMI. Selected V components, that refer to specific cuts of the multidimensional potential energy surfaces, are useful to especially show the dependence on the intermolecular distance R of the total potential v , defined as $V = V + v$, where the centrifugal potential V is calculated using quantized ℓ values.”*.

Reviewer #2 – addressed point 5: *“The authors state that “Their formulation [...] permits to control such prototype CHEMI at increasing complexity of the M reagent.” It is unclear how this control should be achieved.”*

Authors reply and made modifications: We agree with the reviewer#2 and, in order to be more clear, we modified the following sentences at page 7, line 9 from the bottom of the revised manuscript: *“Their formulation, including angular and radial dependences for both real and imaginary components, permits to control the stereo-dynamics of such prototype CHEMI at increasing complexity of the M reagent, passing from an atom to a polyatomic molecule. More specifically, the proposed optical potentials, reproducing most of experimental findings, obtained in our and in other laboratories, are here exploited to cast light on the basic points of general interest for the reaction dynamics stressed above”*.

Reviewer #2 – addressed point 6: *“The authors state that the cross sections obtained from their calculations as well as their energy dependence are consistent with previous experimental results (from Perugia, Lausanne, Eindhoven). However, the results from previous work are not shown (e.g. in a table or in one of the figures given in the article). Hence the reader would have to consult the literature for this. The authors should add the experimental results for comparison.”*

Authors reply and made modifications: We thanks the reviewer#2 for his proper suggestion. To address this point, we modified (with the addition of a new sentence in red colour below with the related new ref. 41) the following text at page 8, line 10 from the bottom of the revised text: *“For the three CHEMI, plotted results, with their different absolute value and dependence on E_{coll} , are consistent with experimental results of Perugia,^{8,39} Lausanne,³⁷ Eindhoven⁴⁰ and Pittsburgh⁴¹ experiments. In*

particular, for the comparison with Perugia experiments see Figure 3 of Ref. 8 and Figure 4 of Ref. 34, with Eindhoven experiments see again Figure 4 in Ref. 34, with Pittsburgh experiments see Figure 4 in Ref. 25, and, finally, with Lausanne experiments see comments to Figure 5 in Ref. 35. Obtained results suggest...". The added new ref. 41 is the following:

41. Gregor, R. W.; Siska, P. E. Differential elastic scattering of $\text{Ne}^*(3s3p2,0)$ by Ar, Kr, and Xe: Optical potentials and their orbital interpretation. *J. Chem. Phys.* **1981**, *74*, 1078-1092.

Reviewer #2 - addressed point 7: "The description of the timescales for collision and rotation is quite lengthy given that it is only used for a five-line-short comparison with experimental results. It also very difficult to link the descriptions for the collision and rotation timescales with the explanation for the energy dependence of the $\text{Ne}^* + \text{NH}_3$ reaction given thereafter."

Authors reply and made modifications: We agree with the reviewer as the old text relating to this point was not clear in our manuscript. To clarify this crucial point, we have rewritten the related text at the end of page 18 of the revised manuscript version, as it follows: "In

general, T_{coll} should be evaluated as (2)

$$T_{coll} = \int_{R_{in}}^{R_c} \frac{dR}{v_r}$$

where

$$v_r = \sqrt{2\mu \left[E - \frac{l(l+1)\hbar^2}{2\mu R^2} - V(R) \right]} \quad (3)$$

defines the radial velocity, R_{in} represents an initial reference distance and R_c is the closest approach distance (turning point). In the case of collisions driven by high anisotropic interactions, v_r can undergo the *levelling effect* of $V(R)$ associated to the most attractive configurations. Under such conditions, R_{in} can be taken as the distance where the interaction anisotropy is comparable with E_{rot} and T_{rot} can be substituted by T_{pend} related to the formed libration-pendular states of the collision complex. Therefore, molecules rotationally relaxed, flying at low velocity and driven by strongly anisotropic interactions, in their collisions they are "forced" to assume the most stable configurations, undergoing the *levelling effect* of $V(R)$."

Furthermore, to address this point, we added also the following sentence at page 19, line 14 from the bottom: "Note that in the first case molecules rotationally relaxed in seeded beams (with an average rotational period of some ps) have been used in low collision experiments,³⁷ while in the second case³⁸ higher collision energies are probed with molecules in effusive beams, populating excited rotational levels."

Reviewer #2 - addressed point 8: "I have counted 14 self-citations in this article. Considering that chemi-ionization has been subject to more than 50 years of research, I doubt that all of these are strictly required. Likewise, some references are missing. Examples:

a) In the introduction, the two mechanisms are introduced. The mechanisms have been known for many years, and reference should be given to this previous work, see, e.g. the work by Hotop and Niehaus (*Z. Physik* 228, 68-88, 1969) and references in Siska (*Rev. Mod. Phys.* 65, 337-412, 1993).

b) Recent work by Dulitz et al. (*Phys. Rev. A* 102, 022818, 2020) also reports on the shortcomings of capture calculations for the description of chemi-ionization.

c) "This is confirmed by the observation of orbiting resonances in $He^* + NH_3$ ". There is earlier work on orbiting resonances in chemi-ionization (Henson et al., *Science* 338, 6104, 234-238, 2012) which should be cited."

Authors reply and made modifications: We thanks the reviewer#2 for his suggestion. To address this point, we deleted Nr. 4 self-citations since they were not strictly necessary in the main text (they are moved to SI). They are the followings: 21. Falcinelli, S.; Pirani, F.; Vecchiocattivi, F. The possible role of Penning ionization processes in planetary atmospheres. *Atmosphere* **2015**, *6*(3), 299-317. 29. Falcinelli, S.; Vecchiocattivi, F.; Farrar, J. M.; Brunetti, B. G.; Cavalli, S.; Pirani, F. Stereo dynamical effects and chemi ionization reactions of atmospheric O₂ and N₂ molecules promoted by collisions with Ne*(³P_{2,0}) atoms. *Chem. Phys Lett.* **2021**, *778*, 138813. 32. Brunetti, B.; Vecchiocattivi, F. *Current Topic on Ion Chemistry and Physics*, Ng, C. Y., Baer, T., Powis I., Eds.; John Wiley & Sons Ltd: New York, 1993; pp 359-445. 44. Cernuto, A.; Tosi, P.; Martini, L. M.; Pirani, F.; Ascenzi, D. Experimental investigation of the reaction of helium ions with dimethyl ether: stereodynamics of the dissociative charge exchange process. *Phys. Chem. Chem. Phys.* **2017**, *19*, 19554-19565.

Then, we added the following sentences:

- To address the sub-point (a), at the end of page 3, line 9 from the bottom of the revised manuscript: "The occurrence of the two mechanisms has been suggested several years ago^{28,29} (see also ref. 30 and references therein) but only recently the nature and selectivity of intermolecular forces involved has been completely addressed.²³⁻²⁵" with the addition of a new related ref.²⁸ which is the following: Hotop, H.; Niehaus, A. Reaction of excited atoms and molecules with atoms and molecules. *Z. Physik* **1969**, *228*, 68-88.

- To address the sub-point (b), at the end of page 19 of the revised manuscript: "Some important results, concerning shortcomings of capture calculations by long range dispersion forces have been recently reported for $He^*(^3S, ^1S) + Li$ CHEMI.⁴⁴ The cases investigated in the present study are more complex, since simultaneously involving, as reagents, a "P" atom, with a high electron affinity ionic core, and molecular partners, with a permanent electric multipole. Related long range forces arise from the balance between dispersion and several other anisotropic interaction components." with the addition of a new related ref. 44 which is the following: Dulitz, K.; Sixt, T.; Guan, J.; Grzesiak, J.; Debatin, M.; Stienkemeier, F.; Suppressing of Penning ionization by orbital angular momentum conservation. *Phys. Rev. A* **2020**, *102*, 022818.

- To address the sub-point (c), we have rewritten the end of the Introduction section of the revised manuscript (see page 6, line 4 from the bottom) as it follows: "Such treatment is quantitative for collision events occurring with relative kinetic energy confined in the thermal and hyper-thermal range of values and provides results that, at a semi-quantitative level, are proper to emphasize important selectivity of the scattering also under sub-thermal conditions. The treatment becomes less accurate when resonances, due to quantum effects, are observed, that effectively manifest when light reagents collide at low kinetic energy. This is confirmed by the observation of resonances in $He^*(^3S_1) + NH_3$ CHEMI^{20,36} due to orbiting effects, emphasized by the reduced mass of the system, that indeed disappear in the $Ne^*(^3P) + NH_3$ CHEMI.³⁷" inserting the correct refs. 20, 36 and 37 as requested by the reviewer.

Reviewer #2 – addressed point 9: *“I strongly recommend having the article proofread and edited by a native English speaker to improve readability.”* **Authors reply and made modifications:** we thank the reviewer#2 for his critical comment. We apologize for the not good English style. To improve the quality of the manuscript, we have deeply revised it by the help of a native English colleague.

Reviewer #2 – addressed minor points 10 and 11: *“Minor points: In Fig. 4, it would also be nice to plot the overall ionization widths. Cartoon in Fig. 4: It would be easier to understand if the chemical formulas were below the graphics.”*

Authors reply and made modifications: we thank the reviewer#2 for his suggestions. To address the first point, and to avoid congesting too much Fig. 4, we modified its caption inserting the following sentence: *“The overall ionization width for each investigated system is approximately given by the sum of the two correlated Γ -functions plotted³⁵”*. Furthermore, in order to address the second point, we modified the cartoon in Fig.4 inserting chemical formulas below the graphics.

Responses to the Reviewer #3 comments:

Reviewer #3 – general comment: *“In this manuscript, the authors study the stereodynamics of chemi-ionization reactions between atomic Ne (in the electronic excited 3P state) and Ar, N₂ and NH₃. Based on the intermolecular interaction potentials obtained previously, here the authors carry out molecular dynamics simulations that allowed them to conclude that the outcome of the collision is determined by the properties of the precursor state (the intermolecular complex) which acts as a transition state of this process. In particular, they suggest that the centrifugal barrier determines the relative weight of the direct and indirect mechanisms. In my opinion, the manuscript does not represent a clear advance, and it is not likely to influence thinking in the field so I cannot recommend it for its publication in Communications Chemistry:...”*

Authors reply: We thank the reviewer#3 for his criticism that gave us the stimulus to improve the quality of our work by better highlighting its novelty and importance. However, we do not agree his final judgment which seems to us too ungenerous with our work whose presentation and discussion surely needs some clarifications by us. To address this critical point, we completely revised our manuscript by a deep rewrite taking into account for a more precise description of the semiclassical procedure used to reproduce experimental data from our and other laboratories. In particular, this was achieved by the following review actions:

- the Introduction section has been reorganized in order to highlight novelty and importance of the paper, addressing comments and criticisms from either reviewers#2 and #3. For such a purpose, the following sentences have been added:

- at page 2, line 6 from the bottom of the revised manuscript: *“In particular, the focus is on barrier-less chemi-ionization reactions in order to provide unique-direct information on basic quantities determining the topology of their stereodynamics. The precursor state (PS), formed by collisions of reagents, plays here an important role being coincident with the reaction transition state. All the features of such a state, as well as its structure and stability, are directly or*

indirectly controlled by intermolecular forces operative in each collision complex.”;

- at page 3, line 6 from the bottom of the revised manuscript: “It is relevant to note that because of an external electronic cloud polarization, the PS $[X\cdots M]^*$ formed at short separation distances tends to assume the configuration of an internal ionic core surrounded by a Rydberg electron. Therefore, DM can be considered as a particular case of ion chemistry, triggered by non-resonant charge transfer coupling entrance and exit channels.²⁵”;
- at page 4, line 5 from the bottom of the revised manuscript: “The present study exploits an accurate formulation of the intermolecular interaction potentials, previously obtained^{23-25,33} and suggested by the complex phenomenology of open shell “P” atoms, investigated by advanced experimental and theoretical methods, by the behaviour of ion-neutral systems, coupled by charge transfer, and by the spectroscopic properties of excimers. The obtained formulation has been tested on experimental findings of CHEMI, investigated in our and other laboratories by coupling scattering and spectroscopic techniques.^{7,8,23-25,34,35} In order to emphasize innovative aspects of the reaction stereo-dynamics, we have found convenient to refer to particular geometries of PS (or pre-reactive state), that open specific reaction channels. However, in the analysis and tests on the experimental findings, the full space of the relative geometries of reagents has been considered. In addition, in the case of NH_3 CHEMI the two considered geometries are the most relevant ones promoting, within selected angular cones, the formation of NH_3^+ ionic product in the ground (X) and in the first excited (A) electronic states.^{7,8,35} The acceptance of angular cones is discussed in detail in Ref. 8. Therefore, the selection of individual reaction channels, triggered by specific cuts of the multi-dimensional interaction potential has been useful to properly address...”;
- at page 6, line 4 from the top of the revised manuscript: “Thanks to the above, we are able to perform a first original attempt to point out how the chemical reactivity depends on the angular momentum quantum number of the intermediate collision complex (precursor or pre-reactive state) leading to the reaction. In particular, we are able to clarify the way in which the centrifugal barrier of the colliding reagents strongly acts as a sort of “selector” of the orbital quantum number effective for the reaction. In our knowledge, this is the first attempt to depict the topology of the stereo-dynamics of a chemical reaction by a state-to-state treatment.”;
- at page 6, line 12 from the bottom of the revised manuscript: “Furthermore, obtained results suggest some limitations of the so called “capture models”, usually exploited to describe many other barrier-less processes, like ion–molecule reactions, occurring in interstellar medium, planetary atmospheres and plasmas.”.

- the addition of a new Methods section (at the end of page 20 of the revised manuscript) and a Supplementary Information linked document able also to address the comments made by the reviewer#2 (see the “**Reviewer #2 – addressed point 3**” above).

Reviewer #3 – addressed point 1: “** It seems that the most important part of the results is the interaction potentials that were calculated previously. The results of the dynamic simulations are shown in Figure 5 (those in Fig.1 are restricted to particular initial geometries of NH₃-Ne), but it is not possible to extract from them the contributions of the direct or indirect mechanism.”

Authors reply: We thank the reviewer#3 for his comment that allow us to clarify the sense of dynamic simulations reported in Fig. 5. For this purpose, we added the following sentences:

- at page 4, line 5 from the bottom of the revised manuscript: “The present study exploits an accurate formulation of the intermolecular interaction potentials, previously obtained^{23-25,33} and suggested by the complex phenomenology of open shell “P” atoms, investigated by advanced experimental and theoretical methods, by the behaviour of ion-neutral systems, coupled by charge transfer, and by the spectroscopic properties of excimers. The obtained formulation has been tested on experimental findings of CHEMI, investigated in our and other laboratories by coupling scattering and spectroscopic techniques.^{7,8,23-25,34,35} In order to emphasize innovative aspects of the reaction stereo-dynamics, we have found convenient to refer to particular geometries of PS (or pre-reactive state), that open specific reaction channels. However, in the analysis and tests on the experimental findings, the full space of the relative geometries of reagents has been considered. In addition, in the case of NH₃ CHEMI the two considered geometries are the most relevant ones promoting, within selected angular cones, the formation of NH₃⁺ ionic product in the ground (X) and in the first excited (A) electronic states.^{7,8,35} The acceptance of angular cones is discussed in detail in Ref. 8. Therefore, the selection of individual reaction channels, triggered by specific cuts of the multi-dimensional interaction potential has been useful to properly address...”;
- i) at page 16, line 12 from the bottom of the revised version of the manuscript: “Additional information is suggested by the Figure 5, especially if analyzed together with the results in Figures 2, 3 and 4. In particular, the relative contribution of two reaction mechanisms can be obtained by combining the results of the turning points in Figure 5 with the radial dependence DM and IM given in Figure 4.”;
- ii) at page 19, line 10 from the bottom of the revised version of the manuscript: “In conclusion, for CHEMI this study emphasizes the important role of PS features, where the distance value of the closest approach R_c , with its dependence on the orbital angular momentum value, is crucial to define the relative role of DM and IM. This target has been achieved by exploiting a full, proper and internally consistent formulation of the intermolecular interaction in some prototypical systems.”.

Reviewer #3 – addressed point 2: “** The methodology used is not sufficiently explained in the manuscript. In particular, I could not find enough details about how the dynamic simulations were carried out.”

Authors reply: We agree with reviewer#3. We believe we have fully addressed this crucial point with the changes we made to the previous two points above and with the inclusion of the linked Supplementary Information document (see the “**Reviewer #2 – addressed point 3**” above).

Reviewer #3 – addressed point 3: “** The authors state that around the closest approach distance the system spends most part of its collision time. However, it seems that trapping in the potential well may play a very important role, in particular for the collisions with NH₃. In fact, according to Fig.5, most of the scattering comes from not significantly large values of *l*. On the same topic, resonances are probably occurring at those energies, leading to the trapping of the intermolecular complex.”

Authors reply: We thank reviewer#3 for his comment which is the same one raised by reviewer#2 (see point 8(c) above). To address this point, we have rewritten the end of the Introduction section of the revised manuscript (see page 6, line 4 from the bottom), as it follows: “Such treatment is quantitative for collision events occurring with relative kinetic energy confined in the thermal and hyper-thermal range of values and provides results that, at a semi-quantitative level, are proper to emphasize important selectivity of the scattering also under sub-thermal conditions. The treatment becomes less accurate when resonances, due to quantum effects, are observed, that effectively manifest when light reagents collide at low kinetic energy. This is confirmed by the observation of resonances in He*(³S₁) + NH₃ CHEMI^{20,36} due to orbiting effects, emphasized by the reduced mass of the system, that indeed disappear in the Ne*(³P) + NH₃ CHEMI.³⁷” inserting the correct refs. 20, 36 and 37 as requested by the reviewer#2.

Reviewer #3 – addressed points 4 and 5: “** I found that some parts of the manuscript are confusing. For example:

** it is not clear which are the components of the imaginary part of the potential that promote the direct and indirect mechanism, and how they were calculated.

** It is not clear why the authors emphasize collisions for Ne + N₂ at theta=0 and not average over all the possible geometries. The same applies for the collisions with NH₃, where it is also not clear why those two initial geometries were highlighted.”

Authors reply: We thank reviewer#3 for his two comments which are shared in part by reviewer#2 (see **Reviewer#2 - addressed point 2** above). To address such points, we added the following sentences at page 4, line 5 from the bottom of the revised manuscript: “The present study exploits an accurate formulation of the intermolecular interaction potentials, previously obtained^{23-25,33} and suggested by the complex phenomenology of open shell “P” atoms, investigated by advanced experimental and theoretical methods, by the behaviour of ion-neutral systems, coupled by charge transfer, and by the spectroscopic properties of excimers. The obtained formulation has been tested on experimental findings of CHEMI, investigated in our and other laboratories by coupling scattering and spectroscopic techniques.^{7,8,23-25,34,35} In order to emphasize innovative aspects of the reaction stereo-dynamics, we have found convenient to refer to particular geometries of PS (or pre-reactive state), that open specific reaction channels. However, in the analysis and tests on the experimental findings, the full space of the relative geometries of reagents has been considered. In addition, in the case of NH₃ CHEMI the two considered geometries are the most relevant ones promoting, within selected angular cones, the formation of NH₃⁺ ionic

product in the ground (X) and in the first excited (A) electronic states.^{7,8,35} The acceptance of angular cones is discussed in detail in Ref. 8. Therefore, the selection of individual reaction channels, triggered by specific cuts of the multi-dimensional interaction potential has been useful to properly address...”.

REVIEWERS' COMMENTS:

Reviewer #2 (Remarks to the Author):

The authors have thoroughly revised the manuscript, taking into account all the reviewers' comments. This has resulted in a much improved version. The article can be published without further changes.

Reviewer #3 (Remarks to the Author):

The points I raised in my report have been reasonably addressed, so I can recommend the manuscript for its publication.

Responses to the Reviewer #2 and #3 comments:

Reviewer #2 – Remarks to the Author: *“The authors have thoroughly revised the manuscript, taking into account all the reviewers' comments. This has resulted in a much improved version. The article can be published without further changes.”*

Reviewer #2 – Remarks to the Author: *“The points I raised in my report have been reasonably addressed, so I can recommend the manuscript for its publication.”*

Authors reply: We are grateful to both reviewers #2 and #3 for their very positive comments and their efforts to help us improve the quality and clarity of our manuscript.